# PRISM: A COMPOSABLE PERSON IMAGE SYNTHESIS MODEL WITH COMPOSITIONAL CONSISTENCY AND UNIFIED OPTIMIZATION

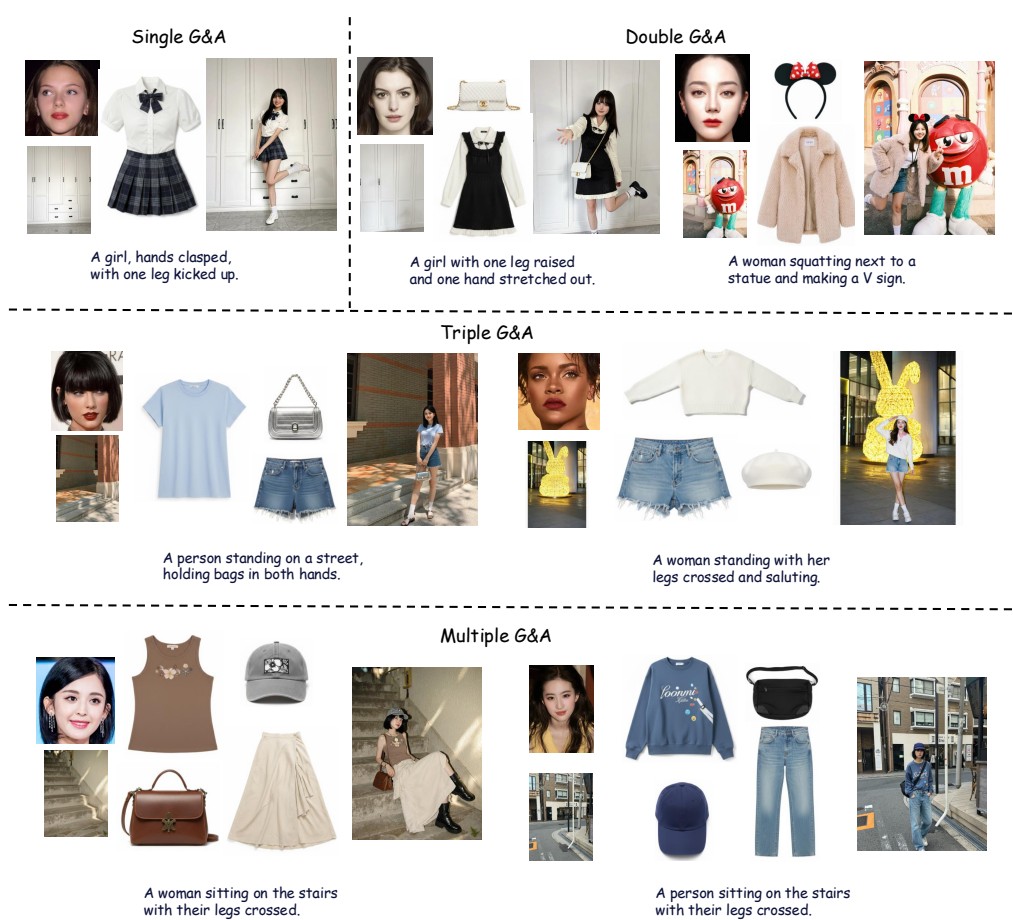

Figure 1: **Showcasing Prism's exceptional ability** to preserve facial identity while faithfully retaining garment, accessory, and background details. The method delivers striking results across scenarios featuring one, two, three, or even more garments and accessories.

## ABSTRACT

While multi-subject reference generation has witnessed rapid advancements, conditional image generation focusing on human-environment interaction, particularly person-centric multi-conditional generation, has received comparatively less attention. This domain encompasses multi-subject referencing, portrait synthesis, and scene guidance. To address this gap, we introduce **Prism**, a unified architecture designed to generate coherent images that satisfy all input conditions, even in the absence of textual prompts. Prism excels at maintaining identity and facial characteristics while aligning with specified backgrounds. Addressing the scarcity of aligned reference and target image sets, we developed a dedicated

pipeline, termed HMS-Dataset, to construct a large-scale training dataset from single images containing individuals. Building upon this, Prism first encodes facial identity, pertinent clothing elements, and background context into sequences. These sequences are subsequently fused via a novel MM-Attention mechanism. Furthermore, we propose a Compositional Consistency Losses (CCL) strategy to incorporate facial similarity, clothing feature preservation, and background consistency, which are specifically designed to boost facial fidelity, retain intricate clothing details, and enhance overall background coherence. Subsequently, guided by the Minimum Variance Distortionless Response criterion, we propose a Unified Gradient Optimization (UGO) update strategy, which enables fair perceptual optimization for multi-objective optimization problems. Ultimately, Prism demonstrates robust identity preservation and seamless human-environment interaction. Evaluated on our proposed PrismBench, Prism achieves state-of-the-art fidelity and controllability, significantly advancing practical applications in character editing and customizable scene synthesis.

# 1 INTRODUCTION

Recent advances in diffusion–based models have reshaped image synthesis, delivering major gains in both fidelity and controllability. Early approaches (Rombach et al., 2022; Ramesh et al., 2022; Song et al., 2020; Sohl-Dickstein et al., 2015; Gal et al., 2022) relied solely on text prompts, and later work (Ye et al., 2023; Wang & Shi, 2023; Ruiz et al., 2023; Zhang et al., 2023; Xu et al., 2023) broadened conditioning to include visual inputs. To meet the demand for higher precision, many methods now provide specialized controls, such as identity preservation (ID) (Li et al., 2024; Wang et al., 2024b; Ye et al., 2025; Yuan et al., 2025), pose transfer (Zhang et al., 2023), depth map guidance (Zhang et al., 2023), and stylization (Hertz et al., 2024). As models have grown in scale and capability, subject-driven generation using single or multiple reference objects has become a central focus. At the same time, practical applications such as virtual try-on (VTON) are emerging, bringing generative AI closer to real-world deployment.

However, far less attention has been paid to person-centric multi-conditional generation, the problem of producing one coherent picture that simultaneously obeys several heterogeneous visual constraints coming from different parts of an image. Typical scenarios involve synthesizing a complete figure that fuses a target identity with several referenced garments and companions and then placing that person into a user-provided background under consistent lighting and perspective while faithfully preserving facial details.

Satisfying all of these requirements in a single forward pass proves difficult for two main reasons. First, the supervision is inherently compositional: preserving facial identity limits pose freedom, while enforcing background consistency restricts how the foreground can be synthesized. Second, the community lacks a public dataset that contains perfectly aligned triplets of face, clothing, and scene at a scale large enough to train diffusion models from scratch. Existing pipelines therefore lean on prompt engineering, template-specific fine-tuning, or iterative inpainting, each of which tends to introduce identity drift, geometric artifacts, or mismatched color tones.

To address these limitations, we introduce **_Prism_**, a unified architecture for person-centric multiconditional generation. Given a face image, target garment images, and a scenic background, Prism seamlessly fuses these inputs into a coherent image that faithfully respects all specified elements. To tackle the scarcity of aligned reference-target pairs, we construct a comprehensive data pipeline that builds a large-scale dataset from single-person images. Moreover, we design a set of specialized loss functions to explicitly enhance facial similarity and preserve fine-grained garment details. Extensive experiments demonstrate that Prism outperforms state-of-the-art methods, including the closed-source models Nano Banana (Google, 2025) and SeedDream 4.0 (ByteDance, 2025), across both quantitative and qualitative evaluations.

Our main contributions are:

- We formulate person-centric multi-conditional generation, highlighting challenges in identity, clothing, and background coherence.

- We build HMS-Dataset and propose **Prism**, featuring MM-Attention, Compositional Consistency Losses, and Unified Gradient Optimization.

- The performance of our proposed PrismBench demonstrates state-of-the-art performance in fidelity, controllability, and human-environment alignment.

## 2 RELATED WORK

**Diffusion Model.** Diffusion has become the dominant text-to-image paradigm, evolving from text-conditioned systems with strong language encoders and classifier-free guidance (Ho & Salimans, 2021) to efficient latent-space models with high fidelity and broad controllability (Esser et al., 2024; Ramesh et al., 2022; Podell et al., 2023; Feng et al., 2023; Balaji et al., 2022). Controllability, specialization, and personalization are advanced by external conditioning branches and parameter-efficient tuning (e.g., LoRA), enabling structural guidance, rapid domain adaptation, and image-conditioned generation via adapters or attention controls (Zhang et al., 2023; Mou et al., 2024; Hu et al., 2022; Gal et al., 2022; Ruiz et al., 2023; Ye et al., 2023; Chen et al., 2024; Cao et al., 2023), motivating unified, instruction-driven pipelines (Xiao et al., 2025; Wu et al., 2025a; Chen et al., 2025b). A concurrent shift replaces convolutional U-Nets with transformer backbones operating over tokenized latents and text, improving global reasoning and scalability through full-sequence attention and adaptive normalization (Peebles & Xie, 2023; Chen et al., 2023). Treating all inputs as tokens in a shared attention space jointly models intra-image structure and cross-modal interactions, and naturally supports multi-subject customization by injecting reference-image token groups with simple positional/type embeddings, masking, or gating—enabling consistent multi-identity scene assembly (Xiao et al., 2025).

**Subject-Driven Image Generation.** seeks to synthesize customized images that faithfully preserve the unique identity and attributes of a given reference subject, effectively bridging the gap between purely text-driven and purely image-driven synthesis. Early methods in this area have split into two principal directions. Tuning-based approaches (Ruiz et al., 2023; Gal et al., 2022; Gu et al., 2023; Kumari et al., 2023) deliver strong fidelity but typically require multiple reference images and additional conditions like a unique identifier. In contrast, tuning-free methods (Li et al., 2024; 2023; Kim et al., 2024; Huang et al., 2025) offer greater convenience and flexibility, yet they often struggle with identity preserving and maintaining consistent subject features across varied prompts and contexts. Recent advancements (Xie et al., 2023; Chen et al., 2025a; Xiao et al., 2025; Mou et al., 2025; Wu et al., 2025b) in subject-driven generation have focused on enhancing architectural components to improve subject consistency, particularly in multi-subject contexts. OmniControl (Xiao et al., 2025) leverages the generative model itself as a reference encoder, while others have introduced novel mechanisms such as systematic data generation pipelines (Wu et al., 2025b), attention-focusing routers (Mou et al., 2025), and token-specific modulation via text streams (Chen et al., 2025a). Despite progress, composable person synthesis remains underexplored yet central in practice (e.g., Nano Banana, Seedream 4.0). We introduce PRISM, a dedicated framework combining robust identity preservation with seamless, context-consistent background integration.

## 3 METHODOLOGY

### 3.1 PRELIMINARIES

In Diffusion Transformers (DiTs) (Peebles & Xie, 2023), multi-modal self-attention integrates image and text tokens. At each step, the latent image $z_t$ is patchified into tokens $X \in \mathbb{R}^{N \times d}$, while the text encoder provides $C_T \in \mathbb{R}^{M \times d}$. After positional encodings (e.g., *RoPE ()*), the concatenated sequence $[X; C_T]$ is processed by multi-head self-attention, enabling bidirectional information flow between visual and textual tokens at every layer:

$$\text{Attn}([X; C_T]) = \text{softmax}\left(\frac{QK^\top}{\sqrt{d}}\right) V \tag{1}$$

Training adopts straight-line flow matching. Intermediate states are defined by

$$z_t = t\, z_0 + (1-t)\, \epsilon, \quad t \in [0,1], \ \epsilon \sim \mathcal{N}(0, I), \tag{2}$$

Figure 2: **Data Curation Pipeline.** Structured conditions—face, garments, and background—are extracted using InsightFace (Deng et al., 2019), Qwen-VL-Max (Wang et al., 2024a), and inpainting-based background reconstruction.

with target velocity $v^\star = z_0 - \epsilon$. The network regresses $v_\theta(z_t, t, C_T)$ using mean-squared error:

$$\mathcal{L}_{\text{diff}} = \mathbb{E}\big[\| v^\star - v_\theta(z_t, t, C_T) \|_2^2\big], \tag{3}$$

## 3.2 HMS-DATASET

**Overview.** To support composable human generation, we introduce **HMS-Dataset**, a high-quality dataset featuring disentangled and multimodal conditions, including face identity, garments, accessories, background, and pose. Unlike existing datasets that lack such fine-grained annotations, HMS is built through an automated and scalable pipeline applied to real-world human images, ensuring both fidelity and diversity.

**Conditions Extraction.** We design a structured pipeline to extract diverse conditioning signals: **(1) Face ID.** Instead of cropping faces from the target image—which may cause overfitting—we retrieve face crops from other images of the same identity (Fig. 2(a)), enhancing generalization and identity robustness. **(2) Garments & Accessories.** Qwen-VL-Max is used to detect and describe visible items (type, color, position), which are then fed into Kontext to generate clean, catalog-style images per item (Fig. 2(b)), enabling disentangled control. **(3) Background.** Kontext, guided by a background-specific prompt, removes humans while preserving scene semantics, offering clean, realistic background conditions.

**Data Filtering.** To ensure high-quality supervision signals, we filter out low-quality samples based on facial clarity, pose correctness, and the success of garment and background generation. Duplicate identities are eliminated through face clustering using embedding similarity. This pipeline results in the **HMS-Dataset**, which contains approximately *1 million* composable samples with diverse and well-aligned conditions, offering clean identities, realistic garments, and semantically consistent backgrounds suitable for conditional generation tasks.

## 3.3 PRISM

In this work, we propose **Prism**, a flexible and unified framework for human image generation with multiple reference images and background guidance, as illustrated in Fig. 3. Prism first encodes multiple input conditions—*e.g.*, background, facial identity, and garments—using a VAE-based (**?**) architecture. To enforce subject-level consistency across modalities, we incorporate a multi-modal attention mechanism inspired by MM-Attention (Xie et al., 2023).

Prism further integrates two core components for effective optimization: (1) **Compositional Consistency Modeling (CCM)**, and (2) **Unified Gradient Optimization (UGO)**. The former explicitly enforces faithful preservation of facial identity, precise reconstruction of garments and accessories, and accurate alignment with the background, each guided by its corresponding reference input.

Figure 3: An overview of the **Prism** framework. (a) Model Structure: Multi-condition references—including face, garments, and background- are encoded via a VAE, with subject consistency injected through MM-Attention and LoRA modules. (b) Compositional Consistency Loss: Identity preservation, garment/accessory fidelity, and background alignment are supervised by $\mathcal{L}_{\text{FaceID}}$, $\mathcal{L}_{\text{G\&A}}$, and $\mathcal{L}_{\text{BCA}}$, respectively.(c) Unified Gradient Optimization: Multi-objective gradients are harmonized under the MVDR projection criterion to ensure fair and stable parameter updates.

The latter balances these multi-objective constraints under the Minimum Variance Distortionless Response (MVDR) criterion, enabling fair and stable parameter updates across diverse loss signals.

### 3.3.1 COMPOSITIONAL CONSISTENCY LOSSES

We define **Compositional Consistency Losses** comprising three task-specific objectives: face identity loss (FIDL), garments and accessories loss (G&AL), and background correspondence attention loss (BCAL), each enforcing consistency over critical aspects of the generated person.

**FIDL: Face Identity Loss.** To preserve the generated character's facial identity, we employ a cosine-based perceptual loss between embeddings of the generated face $\mathbf{z}_{\text{gen}} \in \mathbb{R}^d$ and reference identity $\mathbf{z}_{\text{tgt}} \in \mathbb{R}^d$, extracted via a pre-trained ArcFace backbone (Deng et al., 2019):

$$\mathcal{L}_{\text{base\_ID}} = 1 - \frac{\mathbf{z}_{\text{gen}} \cdot \mathbf{z}_{\text{tgt}}}{\|\mathbf{z}_{\text{gen}}\|_2 \cdot \|\mathbf{z}_{\text{tgt}}\|_2}. \tag{4}$$

This baseline uniformly penalizes all samples, which may over-penalize challenging cases (e.g., occluded or extreme-pose faces). To account for generation difficulty, we introduce an adaptive margin inspired by AdaFace (Kim et al., 2022), using the L2 norm of $\mathbf{z}_{\text{gen}}$ as a quality proxy:

$$\hat{q}_{\text{gen}} = \text{sg}\left(\text{clip}\left(\frac{\|\mathbf{z}_{\text{gen}}\|_2 - \mu_{\|\mathbf{z}\|}}{\sigma_{\|\mathbf{z}\|}/h}, -1, 1\right)\right), \tag{5}$$

where $\mu_{\|\mathbf{z}\|}$ and $\sigma_{\|\mathbf{z}\|}$ are EMA-stabilized batch statistics, $h$ controls concentration, and $\text{sg}(\cdot)$ stops gradients. The final adaptive identity loss is:

$$\mathcal{L}_{\text{FaceID}} = 1 - \left(\frac{\mathbf{z}_{\text{gen}} \cdot \mathbf{z}_{\text{tgt}}}{\|\mathbf{z}_{\text{gen}}\|_2 \cdot \|\mathbf{z}_{\text{tgt}}\|_2} - m \cdot \hat{q}_{\text{gen}}\right), \tag{6}$$

with $m$ controlling the maximum adaptive margin.

**G&AL: Garments and Accessories Loss.** Garments and accessories contain high-frequency details (textures, patterns, logos) that can be blurred by standard diffusion losses. To preserve local structure, we define a center-of-attention coordinate map $\mathbf{F}_{\text{garment}}$ for each query token $q_i$ at $(x, y)$:

$$\mathbf{F}_{\text{garment},(x,y)} = \sum_{k=1}^{N_{\text{garment}}} A_{\text{garment}\to\text{tgt}}[i, k] \cdot \mathbf{G}_k, \tag{7}$$

where $A_{\text{garment}\rightarrow\text{tgt}}$ is the attention matrix and $\mathbf{G}_k$ are normalized reference coordinates. Smoothness is enforced via total variation within the garment mask $\mathbf{M}_{\text{garment}}$:

$$\mathcal{L}_{\text{G\&AL}} = \|\nabla(\mathbf{F}_{\text{garment}} \odot \mathbf{M}_{\text{garment}})\|_1. \tag{8}$$

This encourages locally coherent attention, transferring textures and patterns accurately.

**BCAL: Background Correspondence Attention Loss.** Maintaining background consistency is important for seamless composition. Instead of dense supervision, we use sparse semantic correspondences $\mathcal{C}_{\text{bg}} = \{(u_j, v_j)\}$ between the reference background and target latent:

$$\mathcal{L}_{\text{BCAL}} = -\frac{1}{P_{\text{bg}}} \sum_{j=1}^{P_{\text{bg}}} \log A_{\text{bg}\rightarrow\text{tgt}}[u_j, v_j]. \tag{9}$$

This anchors key background points while allowing the model to plausibly fill in intermediate regions, balancing fidelity and generative freedom.

### 3.3.2 UNIFIED GRADIENT OPTIMIZATION (UGO)

We formulate the training process as a multi-objective optimization problem, aiming to jointly minimize a vector of four task-specific objectives:

$$\min_{\theta} \mathbf{L}(\theta) = [\mathcal{L}_{\text{diff}}(\theta),\ \mathcal{L}_{\text{FaceID}}(\theta),\ \mathcal{L}_{\text{G\&AL}}(\theta),\ \mathcal{L}_{\text{BCAL}}(\theta)]^{\top} \tag{10}$$

Here, $\mathcal{L}_{\text{diff}}$ denotes the standard denoising loss (see Eq. 3), while $\mathcal{L}_{\text{FaceID}}$, $\mathcal{L}_{\text{G\&AL}}$, and $\mathcal{L}_{\text{BCAL}}$ (Eqs. 6-9) serve as auxiliary terms enforcing identity preservation, geometric alignment, and background consistency, respectively. Rather than minimizing a scalarized combination of these losses—which typically requires manual weight tuning-we seek a Pareto-optimal solution, where no objective can be improved without degrading at least one other. This requires a principled mechanism to resolve potentially conflicting gradient directions.

To this end, we adopt a subspace projection approach inspired by the Minimum Variance Distortionless Response (MVDR) principle (Wolfel & McDonough, 2005). Specifically, we constrain the final update direction $\mathbf{g}$ to lie within the subspace spanned by the individual task gradients $\{\mathbf{g}_i = \nabla_\theta \mathcal{L}_i\}_{i=1}^{4}$. The optimal direction is obtained by solving the constrained optimization:

$$\min_{\mathbf{g}=G\boldsymbol{\alpha}} \|\mathbf{g}\|^2, \quad \text{s.t. } \mathbf{v}^{\top}\mathbf{g} = 1 \tag{11}$$

where $G = [\mathbf{g}_1, \ldots, \mathbf{g}_4]$ is the task gradient matrix, and $\mathbf{v}$ is a target direction (e.g., normalized mean gradient). This problem admits a closed-form solution:

$$\boldsymbol{\alpha} = (G^{\top}G + \delta I)^{-1} G^{\top}\mathbf{v}, \quad \mathbf{g} = \frac{G\boldsymbol{\alpha}}{\mathbf{v}^{\top}G\boldsymbol{\alpha}} \tag{12}$$

where $\delta$ is a small regularization constant for numerical stability. The resulting $\mathbf{g}$ yields the minimum-norm update aligned with the target direction, ensuring stable and balanced optimization across tasks. Since $G^{\top}G$ is a $4 \times 4$ system, the computation is lightweight and efficient.

After obtaining the projected update $\mathbf{g}$, we directly apply it to the model parameters. Instead of backpropagating through a weighted sum of losses, we manually overwrite the '.grad' fields with $\mathbf{g}$—reshaped and mapped to each parameter tensor—and perform standard gradient descent:

$$\theta \leftarrow \theta - \eta\mathbf{g} \tag{13}$$

where $\eta$ is the learning rate. This strategy decouples gradient aggregation from loss formulation, enabling more principled and flexible multi-objective training. It integrates seamlessly with common optimizers such as SGD or Adam, and ensures each step follows a Pareto-improving direction under linearly independent task gradients—thus satisfying key assumptions for convergence in multi-objective optimization.

## 4 EXPERIMENTS

### 4.1 EXPERIMENTAL SETTINGS

**Implementation Details.** Building on the Diffusion Transformer (DiT) architecture (Peebles & Xie, 2023) and adopting FLUX.1 dev (Labs et al., 2025) as the backbone, we propose Prism for

Table 1: Quantitative comparison of composable person image generation on **PrismBench**. Gray rows indicate *closed-source models* evaluated via official interfaces.

| Method | FaceSim ↑ | G&ASim ↑ | BackSim ↑ | LPIPS ↓ | Sem. Cons. ↑ |
|---|---|---|---|---|---|
| UNO (Wu et al., 2025b) | 40.52 | 79.78 | 59.71 | 70.16 | 33.32 |
| DreamO (Mou et al., 2025) | 52.42 | 79.67 | 57.44 | 72.37 | 34.29 |
| OmniGen (Xiao et al., 2025) | 51.30 | 84.27 | 65.86 | 67.20 | 33.07 |
| OmniGen2 (Wu et al., 2025a) | 52.86 | 83.05 | 67.92 | 66.06 | 34.28 |
| Xverse (Chen et al., 2025a) | 43.59 | 78.11 | 55.18 | 71.82 | 33.02 |
| SeedDream 4.0 (closed-source) | 55.06 | 81.39 | 79.51 | 37.81 | 34.20 |
| Nano Banana (closed-source) | 54.15 | 84.52 | 81.35 | 35.62 | **34.62** |
| **Prism (Ours)** | **55.27** | **89.32** | **84.38** | **32.51** | 34.52 |

composable person-image synthesis. We equip the network with LoRA adapters (Hu et al., 2022) of rank 128 and an alpha value of 64. The model is optimized with AdamW (Kingma, 2015) at a learning rate of 1e-4, using a batch size of 4 per GPU and training for a fixed number of steps. All training images are resized to $1024 \times 768$, and the experiments are run on 64 H20 GPUs.

**Evaluation Details.** Existing benchmarks for controlled image generation primarily target object-centric synthesis or virtual try-on tasks, lacking datasets tailored to composable person image synthesis with fine-grained control. To address this gap, we introduce **PrismBench**, curated from person-centric subsets of DreamBench (Ruiz et al., 2023) and XVerseBench (Chen et al., 2025a), and extended with additional high-quality portraits.PrismBench comprises 30 identities of diverse ethnicities, with canonical portraits on clean backgrounds, and includes varied apparel types (single to layered garments), accessories, and backgrounds (from simple to cluttered). Each composite image is paired with a background-only counterpart for disentanglement evaluation. The benchmark supports comprehensive evaluation using identity similarity (Face ID, ReID), condition consistency (CLIP-I, CLIP-AS), background preservation (LPIPS, DINOv2), and text–image alignment (CLIP-T). We compare our method with recent multi-subject guided generation models, including MIP-Adapter (Zhong et al., 2025), UNO (Wu et al., 2025b), Dream-O (Mou et al., 2025), Omni-Gen (Xiao et al., 2025), OmniGen2 (Wu et al., 2025a), and Xverse (Chen et al., 2025a). In addition, we evaluate against two *closed-source models*, SeedDream 4.0 (ByteDance, 2025) and Nano Banana (Google, 2025), via their official interfaces for fair comparison. To further complement quantitative evaluation, we conduct user studies involving 30 participants across 100 randomly sampled cases.

## 4.2 MAIN RESULTS

**Quantitative Results.** Table 1 presents a comprehensive comparison of controlled image generation methods on **PrismBench**. **Prism** consistently surpasses both open-source and closed-source baselines across all metrics. In terms of facial identity preservation, Prism achieves the highest **FaceSim** score (55.27), marginally outperforming the strongest baseline (SeedDream 4.0, 55.06). For garment and accessory consistency (**G&ASim**), Prism reaches 89.32, significantly surpassing the next-best (Nano Banana, 84.52). On **BackSim**, which evaluates background alignment, Prism achieves 84.38, indicating a clear advantage over OmniGen2 (67.92) and all others. Prism also yields the lowest **LPIPS** (32.51), demonstrating superior perceptual quality. While **Semantic Consistency** remains competitive (34.52), these results collectively confirm Prism's strength in jointly preserving identity, apparel details, and scene structure—crucial for composable person image synthesis. Notably, even when compared with closed-source models accessed via their official interfaces (gray rows), Prism sets a new state-of-the-art across the board.

**Qualitative Results.** Fig. 4 presents qualitative comparisons between our method and existing approaches such as DreamO, UNO, and Xverse. Compared to these baselines, Prism demonstrates substantial improvements in both facial similarity and garment consistency, while also producing more natural background integration and pose interactions. Notably, our method consistently preserves background coherence across all cases, ensuring spatial continuity even under complex compositional settings. In multi-object interaction scenarios, such as those illustrated in the fourth and

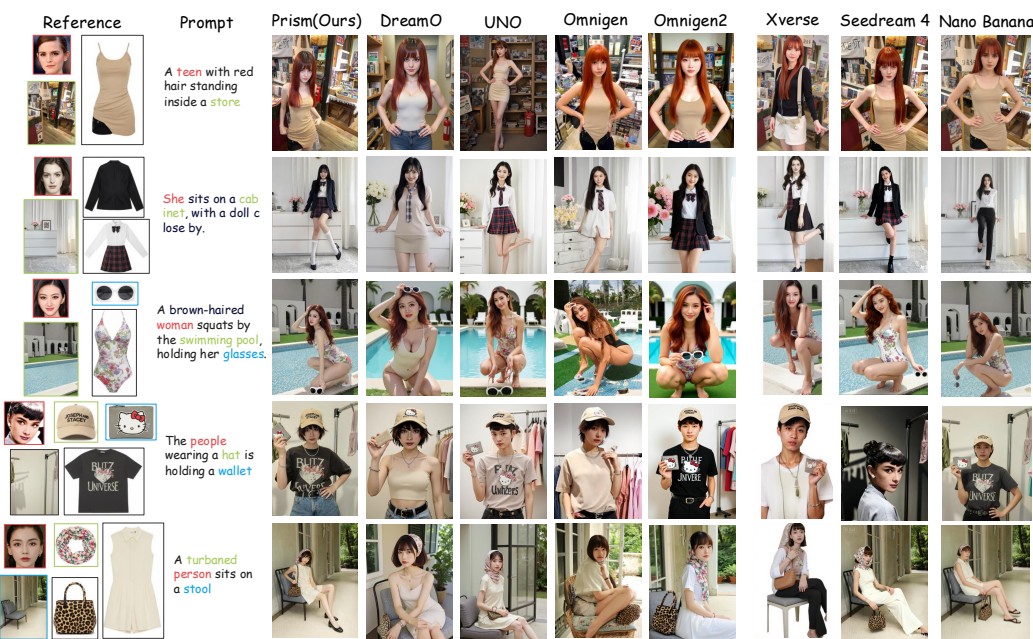

Figure 4: Qualitative Comparison. We compare with Nano Banana and SeedDream 4.0 using their official interfaces, enabling a fair evaluation against closed-source models.

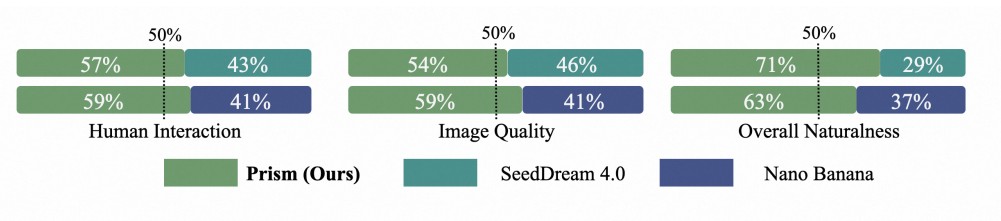

Figure 5: User study on our models with Nano Banana and SeedDream 4.0.

fifth rows, Prism effectively maintains visual consistency between subjects and their accessories. Overall, Prism achieves high fidelity in identity preservation, apparel composition, and coherent integration with the surrounding environment.

**User Studies.** We conduct pairwise user preference studies against SeedDream 4.0 and Nano Banana across three dimensions: *Human Interaction*, *Image Quality*, and *Overall Naturalness*. As shown in Fig. 5, **Prism** consistently outperforms both baselines, with notable advantages in naturalness (71% vs. SeedDream, 63% vs. Nano Banana) and interaction fidelity. These results validate the perceptual quality and controllability of our proposed Prism.

### 4.3 ABLATION STUDIES

**Impact of Compositional Consistency Objectives.** As shown in Table 2, each objective contributes substantially to the overall performance. Removing FIDL leads to a clear drop in facial similarity (49.34 vs. 55.27), while excluding G&AL compromises garment alignment (77.96 vs. 89.32) and degrades visual quality (LPIPS: 57.81 vs. 48.55). Eliminating BCAL significantly impacts background coherence (78.51 vs. 84.38) and weakens semantic alignment (29.56 vs. 34.52). The full objective formulation achieves the best results across all dimensions, confirming the necessity of jointly modeling facial identity, apparel, and scene consistency.

**Impact of Unified Gradient Optimization.** As reported in Table 3, introducing UGO consistently improves all evaluation metrics. It enhances identity retention (55.27 vs. 49.65), improves compositional alignment in clothing (89.32 vs. 78.74) and background (84.38 vs. 71.91), reduces perceptual

Table 2: Impact of each loss terms on the performance of **Prism**, e.g., FIDL, G&AL, and BCAL.

| Method | FaceSim ↑ | G&ASim ↑ | BackSim ↑ | LPIPS ↓ | Sem. Cons. ↑ |
|---|---|---|---|---|---|
| w/o FIDL | 49.34 | 88.24 | 84.21 | 52.77 | **34.57** |
| w/o G&AL | 55.12 | 77.96 | 83.71 | 57.81 | 31.75 |
| w/o BCAL | 54.91 | 88.16 | 78.51 | 68.83 | 29.56 |
| **Prism (Ours)** | **55.27** | **89.32** | **84.38** | **48.55** | 34.52 |

Table 3: Impact of UGO.

| Method | FaceSim ↑ | G&ASim ↑ | BackSim ↑ | LPIPS ↓ | Sem. Cons. ↑ |
|---|---|---|---|---|---|
| w/o UGO | 49.65 | 78.74 | 71.91 | 69.72 | 32.78 |
| w/ UGO | **55.27** | **89.32** | **84.38** | **48.55** | **34.52** |

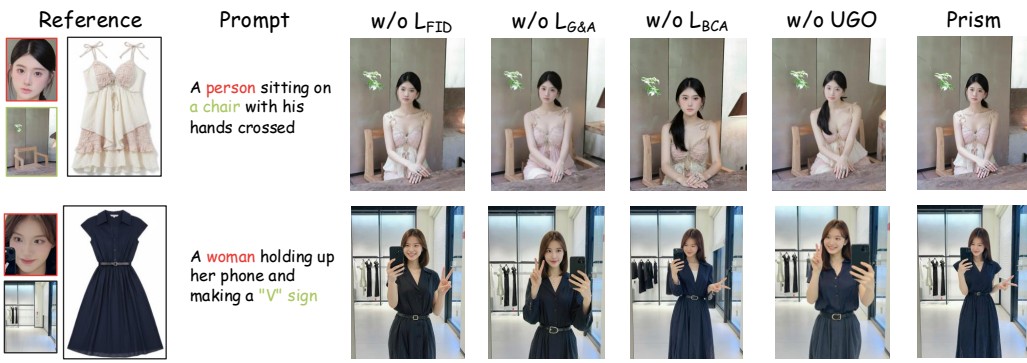

Figure 6: Qualitative results on different losses and UGO modules.

discrepancy (LPIPS: 48.55 vs. 69.72), and boosts semantic correspondence (34.52 vs. 32.78). These improvements validate the effectiveness of UGO in balancing multi-objective optimization for high-quality compositional generation.

**Qualitative Results of Ablation Components.** We provide visual comparisons in Fig. **??** to further assess the contribution of each component. Each objective visibly enhances its corresponding aspect—e.g., facial supervision improves identity fidelity, while garment alignment is better preserved with clothing-specific guidance. The addition of UGO leads to more coherent and natural image synthesis, along with visibly improved global consistency, highlighting its role in stabilizing multi-objective optimization.

## 5 CONCLUSION

We present **Prism**, a unified and composable framework for person-centric image generation under multi-conditional settings, with a particular focus on modeling human-environment interactions. To address the lack of structured supervision, we construct the large-scale HMS-Dataset, enabling fine-grained conditioning from facial identity, garments, and background. Prism integrates a modality-aware attention mechanism for condition fusion and introduces compositional consistency objectives to jointly enforce identity preservation, clothing fidelity, and scene alignment. To further balance the multi-objective optimization process, we develop a Unified Gradient Optimization strategy grounded in the Minimum Variance Distortionless Response (MVDR) principle. Extensive evaluations on the proposed PrismBench demonstrate that Prism achieves state-of-the-art controllability and visual fidelity, offering a scalable solution for character editing and customizable scene synthesis in complex, real-world scenarios.

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

**Algorithm 1** Unified Gradient Optimization (Pseudocode, PyTorch-like)

```python
def UGO_step(model, losses, optimizer, damping=1e-4, weights=None):
    params = [p for p in model.parameters() if p.requires_grad]
    Gs = []
    for L in losses:
        g = torch.autograd.grad(L, params, retain_graph=True, allow_unused=True)
        g = [torch.zeros_like(p) if gi is None else gi for gi, p in zip(g, params)]
        Gs.append(torch.cat([gi.reshape(-1) for gi in g]).detach())

    G = torch.stack(Gs, dim=1) # [D, m]
    m, dev, dt = G.shape[1], G.device, G.dtype

    w = torch.ones(m, device=dev, dtype=dt) / m if weights is None else torch.as_tensor(
        weights, device=dev, dtype=dt)
    w = w / (w.sum() + 1e-12)

    v = (G / G.norm(dim=0).clamp_min(1e-12)) @ w # [D]
    K = G.T @ G + damping * torch.eye(m, device=dev, dtype=dt)
    b = G.T @ v
    alpha = torch.linalg.solve(K, b.unsqueeze(-1)).squeeze(-1)
    g = (G @ alpha) / torch.dot(v, G @ alpha).clamp_min(1e-12)

    optimizer.zero_grad(set_to_none=True)
    offset = 0
    for p in params:
        n = p.numel()
        p.grad = g[offset:offset + n].view_as(p).clone()
        offset += n
    optimizer.step()
```

This supplementary material is organized into several sections, each offering additional details and analysis related to HEAR. The topics covered include:

## A  MORE DETAILS

**Implementation of UGO.** The algorithm is illustrated in Algorithm. 1.

**Limitation.** While Prism demonstrates strong compositional controllability and fidelity, it relies on well-aligned, instance-level reference inputs for each condition (e.g., clean backgrounds, clear facial views, and detailed garments). This dependency may limit its generalization to in-the-wild scenarios where input conditions are noisy, occluded, or incomplete. Additionally, the optimization procedure assumes a fixed set of objectives, which may require adaptation when extending to other compositional dimensions such as lighting, interaction dynamics, or unseen modalities.

**Use of LLMs.** We utilize LLMs to assist with formula derivations and writing refinement.

## B  SUPPLEMENTARY INFORMATION: DATA-CURATION PIPELINE

For completeness, we display the supplementary information that were truncated in Fig. 2, so readers can more readily follow the entire data-curation workflow.

## B.1 GARMENTS & ACCESSORIES EXTRACTION

### Prompt of Garment & Accessory Catalogue Generation

You are an expert fashion analyst and a precise object detection AI. Your primary function is to meticulously analyze the provided image to identify every single garment and accessory worn by the person.

Based on your analysis, you must generate a JSON object that strictly adheres to the following rules:

- **Scope**: Your analysis must be strictly limited to garments (clothing) and accessories. Ignore the person, handheld items (like phones or cups), and any background elements.
- **item_name**: For each item, create a descriptive but concise name. Include its primary color, pattern (if any), and type. For example:

• "shirt" to "white button-up shirt"

• "tie" to "blue and red striped necktie"

• "jeans" to "dark wash denim jeans"

• "glasses" to "black frame eyeglasses"

- **category**: Assign a category to each item. The only two valid categories are:

• Garment: For any piece of clothing (e.g., shirt, pants, jacket, dress, skirt).

• Accessory: For any non-clothing wearable item (e.g., tie, belt, hat, glasses, watch, jewelry, scarf, handbag).

- **total_items_count**: This must be an integer representing the total number of items you have identified in the 'identified_items' array.
- **Output Format**: Your entire response MUST be ONLY the JSON object, without any surrounding text, explanations, or markdown code fences (like "'json).

Now, analyze the image and provide the JSON output.

### Garment & Accessory Catalogue

```
{
  "total_items_count": n,
  "identified_items": [
    {
      "item_name": "white button-up shirt",
      "category": "Garment"
    },
    {
      "item_name": "printed necktie",
      "category": "Accessory"
    },
    ...
  ]
}
```

### Prompt of Garments & Accessories Extract

Extract only the {item_name} from the image. Isolate this item completely, removing the person, the background, and all other items. Recreate it as a clean, flattened studio product shot, as if it's laid perfectly flat and photographed from directly above. The final image must feature only the {item_name}, centered on a solid pure white background.

## B.2 BACKGROUND INPAINTING

> **Prompt of Background Reconstruction Guide**
>
> Seamlessly fill in the background, analyzing the surrounding environment with high precision. Isolate and erase the human figure(s) and all associated objects. Reconstruct the missing background area by meticulously continuing all lines, patterns, and textures from the surrounding context. Pay critical attention to maintaining correct perspective, shadow continuity, and realistic depth of field. The inpainted section must be photorealistically integrated, leaving no visible seams or artifacts.

