# OpenReview forum: "Prism: A Composable Pe\underlinerson Image Synthesis Model with Compositional Consistency and Unified Optimization"
_ICLR.cc/2026/Conference — ICLR 2026 Conference Withdrawn Submission_

### Official Review · Reviewer_jewc · 2025-10-29

**Soundness:** 3
**Presentation:** 2
**Contribution:** 3
**Rating:** 6
**Confidence:** 3

**Summary:**

This paper works on multi-subject conditioned person generation. Specifically, this paper focuses on a virtual try-on setting where a coherent image is expected to generate from a face image, garment images and a background image.
To address this task, this paper presents a data pipeline using large VLM and image generative model to produce a large-scale dataset from single-person image.
To enhance the alignment between generated images and multiple input conditions, authors also design several loss functions to regularize the face identity, garment smoothness and background preservation.
They also present an optimization strategy to harmonize multiple training losses by taking the scale of different losses into consideration.
Experimental results on their presented person-centric PrismBench indicate their finetuned FLUX.1 dev generative model outperform recent open-source and closed source models for multi-concept conditioned generation in jointly preserving identity, apparel details, and scene structure.

**Strengths:**

1. This paper presents strong empirical results on multi-subject–conditioned person generation, with consistent gains in identity preservation, apparel fidelity, and scene consistency.

2. This paper also shows a practical, scalable data pipeline that leverages off-the-shelf large VLMs and image generation/editing models to synthesize diverse training data from single-person images, which is simple to reproduce and effective for multi-subject conditioning.

**Weaknesses:**

1. Many details are missing in the current manuscript. Key components are under-specified, hindering reproducibility: the source(s) of single-image data used for dataset construction; exact evaluation metrics and their computation; how semantic correspondence between the conditional background and generated image is derived for BCAL; and full user-study protocol (participant pool, instructions, interfaces, measures, and statistical tests).
2. How are the proposed losses applied to the denoising framework is unclear.  It is well known that at early steps, the latent would be noisy and make it difficult to estimate a reliable $z_0$. The arcface network used in face identity loss function should be difficult to produce reliable gradients.
3. The manuscript lacks discussion of closely related, contemporaneous methods e.g., [1], and does not clearly position the proposed approach against these in terms of objectives, architecture, and trade-offs.
4. Limited result analysis. It would be interesting to probe performance changes as the number of conditioned subjects increases. Curves or tables stratified by 1/2/3+ subjects would clarify scalability and failure modes in comparison with other models.
5. Several visual results are too small/compressed to judge identity fidelity, garment detail, or background consistency. Higher-resolution figures with consistent layouts and zoom-ins are needed.
6. The visualization is limited. More visualizations are expected in the refinement.
7. In table 2, it would be helpful to also put the baseline results with only standard denoising loss for comparison.

[1] TokenVerse: Versatile Multi-concept Personalization in Token Modulation Space. SIGGRAPH 2025

**Questions:**

The description for Garments and Accessories Loss looks too ambiguous to me. Can authors elaborate how it is motivated and how it is defined?

**Details Of Ethics Concerns:**

In teaser figure, given a latin girl's face the model seems to produce a Western girl or an Asian girl image. This might reflect the bias the generative model in this paper. Plus, all qualitative results in the paper use young girl as the condition. It would be helpful to include more gender and age diversity for visualization.

---

### Official Review · Reviewer_9e34 · 2025-11-01

**Soundness:** 2
**Presentation:** 3
**Contribution:** 2
**Rating:** 4
**Confidence:** 4

**Summary:**

This manuscript presents a unified framework for person-centric image generation with multiple conditions, including identity, garment, accessory, and background. The authors first propose a dedicated pipeline, HMS-Dataset, to construct large-scale data pairs from single input person images. This pipeline utilizes existing image editing models to extract each component from the person's image and filters out invalid samples using a VLM. With this data, the authors train a model conditioned on these multiple inputs and integrate multiple region-level losses. To better balance the losses for each region, the authors also propose a Unified Gradient Optimization method. The experimental results show the competitive performance of the proposed framework compared to existing methods.

**Strengths:**

The proposed data construction pipeline is a clever approach to extracting components from a single person image, which avoids the large labor cost of collecting manually paired data.

**Weaknesses:**

1. The reviewer is not convinced that the "person-centric multi-conditional generation" task, as defined, is new. Many previous works explore this or a highly similar multi-condition person image synthesis task, such as Magic Clothing [1], AnyFit [2], DreamFit [3], AnyDressing [4], IMAGDressing-v1 [5], and Parts2Whole [6]. The proposed model architecture also appears highly similar to the pipelines in these methods. The manuscript fails to adequately differentiate itself from this large body of existing work, so this task definition cannot be recognized as a novel contribution.

2. The dataset is fully synthetic; the inputs (garments, background) are generated by another image editing model (Kontext). Although the extracted components are filtered by VLMs, two problems arise: (1) The AI-generated components may still be inconsistent with the final composed ground-truth image. (2) Since all inputs are purely synthetic, the model may be learning to compose these artifacts rather than real-world, noisy images. The model has not been evaluated on its ability to generalize to real garment or background images, only on the synthetic PrismBench. The authors should evaluate the trained model with more real-world images.

3. The explanation for the proposed "Compositional Consistency Losses" is lacking. There is almost no explanation for the Background Correspondence Attention Loss, and the notation in Eq. (9) is unclear, making it difficult to understand. Moreover, the ablation results (Table 2) show confusing cross-effects. For example, removing the face loss (w/o FIDL) also causes the garment (G&ASim) and background (BackSim) consistency scores to decrease. Why does this happen? If each loss is purely regional and designed to be disentangled, it should primarily affect only its corresponding region. This behavior is not explained.

4. The evaluation is not comprehensive. Given the large number of existing works on this task (as cited in point 1), the authors should have compared against them to truly demonstrate superiority. The current SOTA claim is weak as it only compares against general-purpose multi-subject models, not specialized virtual try-on or person synthesis models.

[1] Magic Clothing: Controllable Garment-Driven Image Synthesis

[2] AnyFit: Controllable Virtual Try-on for Any Combination of Attire Across Any Scenario

[3] DreamFit: Garment-Centric Human Generation via a Lightweight Anything-Dressing Encoder

[4] AnyDressing: Customizable Multi-Garment Virtual Dressing via Latent Diffusion Models

[5] IMAGDressing-v1: Customizable Virtual Dressing

[6] From Parts to Whole: A Unified Reference Framework for Controllable Human Image Generation.

**Questions:**

The manuscript should be further improved by: (1) Clearly stating the novelty and differences compared to the numerous existing methods in this domain. (2) Conducting a more comprehensive comparison against these relevant state-of-the-art methods. (3) Providing a clearer explanation of the loss functions and results in the ablation study.

---

### Official Review · Reviewer_1d9z · 2025-11-01

**Soundness:** 3
**Presentation:** 3
**Contribution:** 4
**Rating:** 6
**Confidence:** 3

**Summary:**

This article proposes Prism, a multi-condition composable character synthesis for "human-environment" : encoding the face, clothing accessories, and background respectively and fusing them through MM-Attention, combined with CCL and UGO to balance the identity, clothing details, and background alignment. Meanwhile, an automated data pipeline, HMS-Dataset, is constructed to support training.

**Strengths:**

1. The method design is unified and pluggable. The proposed MM-Attention,CCL, and UGO have clear logic.
2. The proposed automated data pipeline, HMS-Dataset, is very interesting.
3. On PrismBench, indicators such as face similarity, clothing consistency, background consistency, and LPIPS are generally superior to those of SOTA.

**Weaknesses:**

1. It has a strong reliance on clean and well-aligned reference conditions. In the future, more data can be added to address more scenarios.

2. I'd like to know how the scenario would turn out in the absence of certain conditions? Can it work properly?

**Questions:**

The implementation details show that the training resources are relatively large. How about the costs such as training and inference video memory.

---

### Official Review · Reviewer_tWGZ · 2025-11-03

**Soundness:** 2
**Presentation:** 2
**Contribution:** 2
**Rating:** 4
**Confidence:** 4

**Summary:**

This paper presents PRISM, a framework for composable person image synthesis that aims to integrate multiple conditions (face identity, garments/accessories, background) into a coherent output. The key contributions include: (1) the construction of the HMS-Dataset, a large-scale dataset with disentangled conditions; (2) a set of Compositional Consistency Losses (FIDL, G&AL, BCAL) to enforce fidelity on specific aspects; and (3) a Unified Gradient Optimization (UGO) strategy based on the MVDR criterion to balance these multi-objective losses. The method is evaluated on a newly proposed benchmark, PrismBench, and is shown to outperform several open-source and closed-source baselines quantitatively and qualitatively.

**Strengths:**

The work is extensive, covering dataset creation (HMS-Dataset), a novel model framework with specialized losses, and a new optimization strategy (UGO).

The quantitative results on the proposed PrismBench are impressive, showing state-of-the-art performance across multiple metrics compared to strong baselines, including closed-source models. The qualitative results also appear compelling.

**Weaknesses:**

Lack of Conceptual Novelty: The core technical components feel like a systematic and well-engineered combination of existing ideas rather than a fundamental conceptual breakthrough.

The use of specialized losses for face, garment, and background is a common and intuitive strategy in compositional generation.

The UGO method, while effectively applied, is based on the well-established MVDR beamforming principle from signal processing, adapted here for gradient fusion. Similar gradient manipulation techniques for multi-task learning exist in the literature.

The overall architecture relies on a pre-trained DiT/FLUX backbone with LoRA and attention mechanisms for multi-condition fusion, which is a standard paradigm.

**Questions:**

No Questions

---

### Note · Authors · 2025-12-04

I have read and agree with the venue's withdrawal policy on behalf of myself and my co-authors.